# Risk of Adverse Pregnancy Outcomes for Women with IBD in an Expert IBD Antenatal Clinic

**DOI:** 10.3390/jcm11102919

**Published:** 2022-05-22

**Authors:** Gillian Lever, Hlupekile Chipeta, Tracey Glanville, Christian Selinger

**Affiliations:** 1Leeds Gastroenterology Institute, Leeds Teaching Hospitals NHS Trust, St James University Hospital, Bexley Wing, Leeds LS9 7TF, UK; gillian.lever@nhs.net; 2Leeds Institute of Medical Research at St James’s, University of Leeds, Leeds LS2 9JT, UK; 3Department of Obstetrics, Leeds Teaching Hospitals NHS Trust, St James University Hospital, Leeds LS9 7TF, UK; hlupekile.chipeta@nhs.net (H.C.); tracey.glanville@nhs.net (T.G.)

**Keywords:** inflammatory bowel disease, pregnancy, maternal outcomes, neonatal outcomes

## Abstract

Background: Patients with inflammatory bowel disease (IBD) are at increased risk of adverse outcomes from pregnancy. It is unclear whether IBD indications account for the higher rate of Caesarean section (CS) in IBD patients. Methods: A retrospective cohort study of 179 IBD patients cared for in a dedicated combined IBD antenatal clinic and 31,528 non-IBD patients was performed. The outcomes were method of delivery, preterm birth, birthweight, admission to neonatal intensive care unit (NICU), and stillbirth. We analysed the associations between disease activity, medication with method of delivery, and neonatal outcomes. Results: Delivery by CS was more common in IBD patients (RR 1.45, CI 1.16–1.81, *p* = 0.0021); emergency CS delivery was equally likely (RR 1.26, CI 0.78–2.07, *p* = 0.3). Forty percent of elective CS were performed for IBD indications. Stillbirth was five-fold higher in IBD patients (RR 5.14, CI 1.92–13.75, *p* < 0.001). Preterm delivery, low birthweight, and admission to NICU were not more common in patients with IBD, and IBD medications did not increase these risks. Active disease during pregnancy was not associated with adverse outcomes. Conclusions: Delivery by CS was more frequent in the IBD cohort, and most elective CSs were obstetrically indicated. A dedicated IBD antenatal clinic providing closer monitoring and early detection of potential issues may help improve outcomes.

## 1. Introduction

Patients with inflammatory bowel disease (IBD) are often of child-bearing age, and pregnancy poses additional clinical questions requiring careful IBD and obstetric management [1,2]. Between 300,000 and 500,000 people are thought to suffer from IBD in the United Kingdom. Having a diagnosis of IBD influences patients’ experiences of pregnancy and delivery, and affects outcomes for both the mother and baby.

As disease activity at conception often predicts disease activity during pregnancy, patients with IBD should aim for remission prior to pregnancy [2,3,4,5,6]. In patients with active disease, the risk of adverse maternal and neonatal outcomes increases; thus, careful monitoring, medication optimisation, and input from gastroenterologists are essential for the obstetric care of patients with IBD [2,3,4,5,7,8,9,10,11]. 

Using IBD medication is low risk during pregnancy, and has mostly been absolved of any contribution to adverse neonatal outcomes. Current guidance recommends that only methotrexate, thalidomide, and newer molecules such as tofacitinib or ozanimod should be avoided [2,4]. 

IBD increases the risk of delivery by Caesarean section (CS); however, the full reasons for this increase have yet to be established [12,13]. We recently reported the perineal outcomes in our patient cohort [14]. Specific indications amongst the IBD population exist for elective CS delivery, such as active perianal disease and ileal pouch anal anastomosis (IPAA); however, these indications do not completely explain the higher rate of CS in this group [3,9].

Delivery of preterm and small infants is more commonly observed for mothers with IBD. However, the risk of outcomes such as stillbirth or neonatal intensive care unit (NICU) admission have not been consistently demonstrated in the infants of IBD patients [3,4,7,12,15]. 

In this large, single-centre cohort of pregnant women with IBD, we aimed to examine the indications for delivery by Caesarean section, the risk of adverse infant and maternal outcomes, and the impact of disease activity and medication on these outcomes.

## 2. Materials and Methods

In this retrospective cohort study, we compared maternal and neonatal outcomes of pregnancies in IBD and non-IBD patients in a large secondary and tertiary NHS Trust. Pregnant IBD patients were identified through the Combined IBD Antenatal Clinic (“Combined Clinic”), which started in 2014. The combined clinic is jointly run by a consultant physician with expertise in IBD and a consultant obstetrician with expertise in feto-maternal medicine. All pregnant patients with a diagnosis of IBD who were registered to give birth in Leeds were offered appointments in the combined clinic, therefore more reflecting the local IBD population rather than a referral cohort. All consultations were conducted with both experts present, which allowed true multidisciplinary work and reduced the risk of mixed messages being received by patients. All pregnant patients with IBD were identified through IBD clinics, IBD helplines, and community midwife referrals. IBD patients who were seen in the Combined Clinic and delivered in Leeds Teaching Hospitals between 2014 and April 2018 were included in this study. 

Non-IBD patients included in this study were those who delivered in Leeds Teaching Hospitals between 2015 and April 2018. The later commencement of control data collection is due to very few recorded deliveries from IBD patients in 2014. 

To minimise variation in obstetric care pathways between patients, only those patients who delivered in Leeds Teaching Hospitals were included. All patients with a multiple pregnancy were excluded from this study as the IBD cohort exclusively consisted of singleton pregnancies. Pregnancies that did not reach viability (below 24 weeks’ gestation) were excluded.

In IBD patients, the impact of disease activity, medication, and disease characteristics on maternal and neonatal outcomes were assessed. Disease activity was determined in the Combined Clinic using the Physician’s Global Assessment Score for each trimester. Medication use in IBD patients was analysed based on timing relative to the pregnancy. Patients were classed as “exposed” if prescribed IBD medication (including steroids) during pregnancy. All exposure during pregnancy was reported as one variable, and maintenance medication was continued for at least two trimesters. Patients classed as “unexposed” were either naïve to the specific medication or had not been prescribed it within at least three months prior to and during pregnancy. 

Routinely collected data for all patients were extracted from the K2 Medical Systems™ maternity database. Maternal data included: age, parity, and method of delivery. Neonatal data included: birthweight, gestation, admission to neonatal intensive care unit (NICU), and stillbirth. For IBD patients, additional routine data (subcategories of IBD, disease activity and medication use) were recorded in the Combined Clinic. Maternal and neonatal outcomes were compared between IBD and non-IBD patients. IBD subcategories (Crohn’s disease (CD) and ulcerative colitis (UC) only) were also compared with those of non-IBD patients. 

The method of delivery was recorded as vaginal (spontaneous vaginal and breech vaginal), operative (forceps and ventouse), or Caesarean section (CS). CS delivery was subdivided by urgency into categories one to four. Categories one and two were classed as “emergency Caesarean section”, and were carried out within a specified timeframe due to the immediate risks of maternal or foetal compromise. Category three required expedited delivery, but without concern for maternal or foetal compromise, and category four was a pre-planned CS delivery [16]. We classified categories three and four as “Elective Caesarean Section”, as their timing is at the discretion of the supervising obstetrician [16]. 

The studied infant outcomes were prematurity, birthweight, requirement for admission to a neonatal intensive care unit (NICU), and stillbirth. Low birth weight (LBW) was defined as <2500 g and very low birth weight (VLBW) as <1500 g. Small for gestation age (SGA) was defined as birthweight below the 10th percentile. Prematurity was defined as birth before the 37th week of gestation.

Data were analysed with IBM^®^ SPSS^®^ Statistical Software (IBM Corp. Released 2016. IBM SPSS Statistics for Windows, Version 24.0. Armonk, NY, USA: IBM Corp) and OpenEpi Epidemiological Statistics (Dean AG, Sullivan KM, Soe MM. OpenEpi: Open Source Epidemiologic Statistics for Public Health, Version). Categorical data were examined via chi-squared and relative risk calculations, and numerical data were examined via independent group t-tests. Due to the observational nature of the study, no sample size calculations were performed. Missing values accounted for the difference between reported outcomes and total patients in category.

As a clinical audit of routine clinical data, this study was exempt from Research Ethical approval [17].

## 3. Results

We analysed the data from 31,707 births, 179 of which were from mothers with IBD, and 31,528 were from mothers without IBD (non-IBD). Of the IBD patients, 90 had Crohn’s disease, 79 had Ulcerative Colitis, and 10 had unclassified IBD. 

The parity of mothers was divided into primiparous (first birth) or multiparous (second birth or more). The primiparous mothers accounted for 90 IBD births and 12,549 non-IBD births (Table 1). The IBD cohort had a significantly higher proportion of primiparous mothers than the non-IBD cohort (RR 1.34, CI 1.16–1.54, *p* = 0.0016). IBD patients were, on average, almost two years older than non-IBD patients (31.6 years vs. 29.8 y, CI 1.07–2.74, *p* < 0.001). 

### 3.1. Pregnancy

#### 3.1.1. Disease Activity

A total of 158 IBD patients had their disease activity recorded both prior to and during pregnancy. Pre pregnancy, 142 patients were in remission, 13 of whom had newly active disease during pregnancy. There were 16 patients with active disease pre-pregnancy; 11 of these continued with active disease during pregnancy and five of these went into remission during pregnancy. The risk of active disease during pregnancy was over seven-fold greater in patients with active disease pre-pregnancy (RR 7.51, CI 4.06–13.88, *p* < 0.001). 

#### 3.1.2. Medication

A total of 125 of the 179 IBD patients were exposed to one or more medications during their pregnancy (Table 2). A total of 15% of IBD patients were on not on medication during pregnancy. 

Patients with CD were most commonly exposed to thiopurine (41.11%), patients with UC were most commonly exposed to ASA (67.09%), and patients with IBDU were most commonly exposed to ASA (60%). Steroid use during pregnancy was similar between CD and UC patients (Table 1).

### 3.2. Delivery

#### 3.2.1. Method of Delivery

Delivery via CS was more common in IBD patients than in non-IBD patients (30.17% vs. 20.80%; Table 2), with IBD patients being 1.45 times as likely to deliver this way (RR 1.45, CI 1.16–1.81, *p* = 0.0021) (Table 3). CS delivery was also more likely in IBD patients in the subanalysis of primiparous mothers (33.33% vs. 20.78%, RR 1.61, CI 1.20–2.15, *p* = 0.0035). Spontaneous vaginal delivery was less common in IBD patients than in non-IBD patients (54.19% vs. 64.73%), and the difference between the groups was statistically significant (RR 0.64, CI 0.48–0.87, *p* = 0.0033). The risk of instrumental delivery was no different between IBD and non-IBD patients (15.64% vs. 13.97%, RR 1.12, CI 0.80–1.58, *p* = 0.52).

When analysing Caesarean deliveries, there was no significant difference between the IBD and non-IBD groups in terms of rates of emergency CS compared with elective CS (35.19% vs. 40.65%, RR 0.87, CI 0.60–1.25, *p* = 0.42) or compared with all methods of delivery (10.61% vs. 8.46%, RR 1.26, CI 0.78–2.07, *p* = 0.30). Elective CS was more common in IBD patients than in non-IBD patients (21.88% vs. 13.49%, RR 1.45, CI 1.09–1.95, *p* = 0.02). 

Focussing on IBD subtypes, CS delivery overall remained more common in patients with CD and UC when each group was compared with the non-IBD patients (CD 27.78%, UC 31.65%, non-IBD 20.80%), with only the UC group demonstrating a significantly higher risk (RR 1.51, CI 1.09–2.01, *p* = 0.018). The subanalysis of elective and emergency CS showed that the patients with CD were more likely to undergo elective CS than non-IBD patients (21.69% vs. 13.49%, RR 1.61, CI 1.07–2.42, *p* = 0.029). 

The rates of elective CS in the UC cohort were not significantly different from those of the non-IBD cohort. Emergency CS rates showed no significant difference in either CD or UC groups compared with the non-IBD patients. 

One-quarter of patients with active disease during pregnancy delivered via CS; however, there was no association between disease activity and either CS delivery overall (RR 0.76, CI 0.37–1.58, *p* = 0.45) or emergency CS (RR 0.96, CI 0.31–3.02, *p* = 0.91).

#### 3.2.2. Indications for CS

IBD patients who delivered via CS were split into CS performed for either obstetric or IBD indications (Table 4). Of the 54 CS deliveries in the IBD group, 19 were emergency CS (35.19%) and 35 were elective CS (64.81%). All emergency CS in IBD patients were obstetrically indicated, most commonly for presumed foetal compromise (73.68% of emergency CS) (Table 4). Of the elective CS patients, 60% had an obstetric indication and 40% had an IBD indication. The most common obstetric indications for elective CS in IBD patients were breech positioning (52.38%) and previous CS (28.57%). All IBD indications for CS were relative or absolute based on current guidelines [2,4]; thus, all patients with active perianal disease or IPAA underwent CS delivery. 

One CS performed due to a perforated ileum was still classified as “elective” because, whilst the delivery was expedited (Category 3), there were no immediate concerns about foetal compromise, and the timeframe of delivery was at the discretion of the obstetrician.

### 3.3. Outcomes (Neonatal)

#### 3.3.1. Gestation and Weight

The proportion of preterm deliveries (gestation < 37 weeks) was equal in both IBD and non-IBD cohorts, with no significant difference demonstrated (8.52% vs. 8.47%, RR 1.01, CI 0.62–1.64, *p* = 0.98; Table 5). The mean birthweight of infants born to IBD and non-IBD mothers was similar (3282 g vs. 3328 g, mean difference −45.94 g, CI −131.88–40.01, *p* = 0.30; Table 5). Infants classified as low birthweight (LBW, <2000 g) or very low birthweight (VLBW, <1500 g) occurred at similar rates in both IBD and non-IBD groups (LBW 6.15% vs. 6.90%; VLBW 1.68% vs. 1.16%). No difference in the risk of infants classed as small for gestational age was observed between IBD and non-IBD patients (SGA, birthweight below 10th centile for gestation) (RR 0.9, CI 0.54–1.52, *p* = 0.7). 

#### 3.3.2. NICU Admission and Stillbirth 

The risk of admission to neonatal intensive care (NICU) immediately after birth was similar in infants of IBD mothers and in those of non-IBD mothers (2.96% vs. 3.52%; RR 0.84, CI 0.35–2.00, *p* = 0.69; Table 5). 

Stillborn infants from all patients (IBD and non-IBD) accounted for 141 births. Stillbirth was strongly associated with preterm delivery (RR 25.71, CI 17.55–37.67, *p* < 0.001), LBW (RR 26.71, CI 18.64–38.28, *p* < 0.001,) and VLBW (RR 62.61, CI 45.04–87.06, *p* < 0.001). In the IBD cohort, there were four stillborn infants, which is a significantly higher proportion than in the non-IBD cohort (2.23% vs. 0.44%; RR 5.14, CI 1.92–13.75, *p* < 0.001). In the IBD group, a detailed root cause analysis of the four stillbirths was undertaken by senior obstetric and gastroenterology clinicians who had not been involved in the antenatal care of those patients. Glucose metabolism changes due to steroid therapy may have contributed to one case, whereas for the other three cases, no connection to IBD or treatment was found (Appendix A).

#### 3.3.3. Impact of IBD Medication

Exposure to mesalazine, thiopurine, or biologics during pregnancy was analysed for potential impact on infant outcomes. Patients on each IBD medication were compared with those on either no IBD medication or other IBD medication. When patients on biologics were compared with those taking other IBD medication, the risk of preterm birth was greater in the biologics group (RR 3.29, CI 1.23–8.79, *p* = 0.014); however, there was no increased risk in the biologics group compared with patients on no IBD medication (RR 4.97, CI 0.65–38.11, *p* = 0.075). Neither mesalazine nor thiopurine conveyed any increased risk of adverse infant outcomes compared with either other IBD medication or no IBD medication. Biologics did not confer an increased risk of any adverse outcome, aside from preterm birth, compared with either other IBD medication or no IBD medication (Appendix A Appendix A). 

Exposure to mesalazine or thiopurine during pregnancy compared with no IBD medication was not associated with adverse infant outcomes (Appendix A Appendix A). Disease activity during pregnancy (Appendix A Appendix A) did not correlate with adverse infant outcomes.

## 4. Discussion

In our study, we examined the outcomes of IBD pregnancies in a cohort under an expert, combined IBD antenatal clinic. In this large cohort, we confirmed the increased risk of adverse maternal and foetal outcomes in pregnant women with IBD; however, the overall risk of severe adverse outcomes was low.

In line with the established evidence base, our study confirmed that a diagnosis of IBD confers an increased risk of CS delivery [2,9], and demonstrated that IBD patients were significantly less likely to experience a spontaneous vaginal delivery than non-IBD patients. When analysing IBD subtypes, a previous meta-analysis demonstrated the significance in the risk of CS delivery in CD patients but not than in UC patients when each group was compared with controls [9]. We found that CD only demonstrated a significantly increased risk of elective CS rather than all CS (emergency and elective) compared with non-IBD patients. Moreover, when analysing all CS deliveries, we found that UC, rather than CD, conferred an increased risk of this method of delivery compared with non-IBD patients. This effect could only be partially explained by some women requiring CS for IPAA. Reassuringly, we found that IBD patients had no higher risk of delivery by emergency CS than the non-IBD population. 

Previous studies have not focussed on the indications for CS in IBD patients; thus, increased rates of CS in this group are insufficiently understood. Whilst there are several absolute IBD indications for CS (active perianal disease and IPAA), data on their frequency are lacking. Over half of IBD CS deliveries were performed for obstetric indications. By applying strict criteria for IBD indications for CS (active perianal disease and IPAA patients), only 40% of elective CS were for IBD indications [2,4,18].

Pregnant women with IBD were older than the controls and were more likely to be primiparous. This is in keeping with the evidence that women with IBD tend to have children later in life than healthy controls [1].

IBD increases the risk of adverse outcomes in pregnancy, including prematurity and lower birthweight [9,11]. We found no associations with an increased risk of preterm birth, low or very low birthweight, or infants classed as small for gestational age. All patients were reviewed in combined IBD antenatal clinics with senior obstetric and gastroenterology clinicians. This could have positively impacted outcomes. Patients were monitored more closely with additional foetal growth scans than non-IBD patients, providing a greater opportunity to identify growth retardation or risk factors for preterm delivery. 

Despite the encouraging outcomes for gestation, birthweight, and NICU admission rates, the rate of stillbirth in our IBD cohort was over four-fold higher. In the U.K., the national rate of stillbirth is 0.4%, a figure matched by that in our non-IBD cohort [19]. Only four IBD patients experienced stillbirth compared with 137 non-IBD patients; so, despite statistical significance, the numbers were small. The risk of stillbirth strongly correlated with preterm gestation and lower birthweights in the entire sample; however, in IBD patients, only one stillborn infant was preterm and small, and one infant was preterm but had a normal range birthweight. The stillbirths in the IBD cohort were investigated using root cause analysis, and there was no evidence to suggest that IBD-related complications had contributed to the outcome; however, smoking and potential side effects from steroid therapy could have contributed. Studies on the stillbirth risk in IBD patients herald mixed results, which suggests that the risk is equal to that in the non-IBD population [2,11], but active disease was implicated as a risk factor [4].

We confirmed that disease activity prior to conception correlates with disease activity during pregnancy [2,3,12,13]. Antenatal disease activity did not increase the risk of emergency CS, suggesting that disease activity alone should not be classed as an indication for elective CS in IBD patients. This is also demonstrated by the finding that most patients with active disease in our study had a spontaneous vaginal delivery. 

The reported adverse impact of active disease on pregnancy outcomes was not demonstrated in our study [2,4,5,6,7,9,12,20]. Low rates of active IBD during pregnancy may have mitigated the effects of IBD on pregnancy outcomes. It is possible that the negative effects of active disease on pregnancy outcomes could be found in larger samples, but we reported on a larger cohort than many other IBD studies.

IBD medications (excluding methotrexate and thalidomide) are mainly low risk throughout pregnancy [2,9,21,22]. In our cohort, patients exposed to biologics had an increased risk of preterm delivery, but there were no other significant differences in adverse neonatal outcomes relating to IBD medications. This unadjusted finding may have occurred by chance or may reflect biologics-exposed patients having more severe disease. 

The main strength of our study lies in being a single-centre cohort with consistent IBD care and a uniform approach to obstetric care, which reduces the bias from variation in practice, such as thresholds for CS, instrumental delivery, or the approach to foetal growth scanning compared with multicentre studies. There are also several limitations to our study. There were few patients with active IBD during pregnancy, which may explain why we did not observe an association between active disease and adverse pregnancy outcomes. In keeping with clinical guidelines at the time, we did not routinely assess faecal calprotectin during pregnancy and relied on the more subjective PGA. Smoking data were unfortunately incomplete, and any associations with outcomes could not be analysed. Furthermore, due to variances in care between centres not all of our findings may apply to other centres and health systems. 

## 5. Conclusions

In conclusion, we demonstrated that with the strict application of international guidance, 40% of elective CS are for IBD reasons, which partially explains the higher rate of CS in this group. Obstetric indications for elective CS in IBD patients were most commonly breech position and previous CS. It is possible that patients who had previously delivered by CS were advised to do so in the past due to their IBD. Careful application of international guidance could thus have led to a reduction in the proportion of IBD patients delivering by CS. The finding that emergency CS was equally likely in IBD and non-IBD patients adds more weight to current guidance, which only advises IBD-indicated CS in a limited number of clinical conditions. 

IBD patients were found to be at no increased risk of important adverse neonatal outcomes such as preterm birth, low birthweight, and admission to the NICU. Whilst stillbirth was a significant risk in our IBD cohort, root cause analysis of each case did not implicate IBD as the cause. We were able to confirm the safety of IBD medication during pregnancy, with no significant increased risk of adverse neonatal outcomes in medicated patients. These overall encouraging findings demonstrate the enormous value in having a dedicated combined IBD antenatal clinic providing consistent care in line with current international guidelines.

## Figures and Tables

**Table 1 jcm-11-02919-t001:** Study cohort.

Patient Characteristics	IBD				Non-IBD	*p*
	All IBD	CD	UC	IBDU		
Age (mean)	31.62	31.2	32.06	31.9	29.72	0.000009
Age (median)	32	31	33	33.5	30	
Primiparous, N (%)	90 (50.28)	46 (51.11)	40 (50.63)	4 (40)	12,549 (39.83)	0.0016
Multiparous, N (%)	79 (44.13)	38 (42.22)	36 (45.57)	5 (50)	18,960 (60.18)	
Parity not recoded, N (%)	10 (5.59)	6 (6.67)	3 (3.80)	1 (10)	1108 (3.52)	
TOTAL	179	90	79	10	31,509	

IBD—inflammatory bowel disease; CD—Crohn’s disease; UC—ulcerative colitis; IBDU—inflammatory bowel disease unclassified.

**Table 2 jcm-11-02919-t002:** Medication exposure (“Exposed” patients had been prescribed either IBD medication or steroids during pregnancy. “Unexposed” patients were either naïve to the specific medication, or had not been prescribed it within at least three months prior to pregnancy. Table percentages reflect the proportion of total study IBD population (N = 179) and IBD subtypes (CD N = 90; UC N = 79; IBDU N = 10). Missing data are reflected where totals do not add up to N. ASA—aminosalicylates.

Medication Exposure during Pregnancy		IBD	CD	UC	IBDU
		N (% of All IBD Pts)	N (% of CD Pts)	N (% of UC Pts)	N (% of IBDU Pts)
ASA	Exposed	65 (36.31)	6 (6.66)	53 (67.09)	6 (60)
	Non-exposed	83 (46.37)	64 (71.11)	15 (18.99)	4 (40)
Thiopurine	Exposed	60 (33.52)	37 (41.11)	22 (27.85)	1 (10)
	Non-exposed	88 (49.16)	33 (36.67)	46 (58.23)	9 (90)
Biologic	Exposed	38 (21.22)	26 (28.89)	8 (10.13)	4 (40)
	Non-exposed	141 (78.77)	44 (48.89)	60 (75.95)	6 (60)
Steroids	Exposed	18 (10.06)	8 (8.89)	8 (10.13)	2 (20)
	Non-exposed	72 (40.22)	35 (38.89)	33 (41.77)	4 (40)
	Not recorded	89 (49.72)	47 (52.22)	38 (48.10)	4 (40)
Any IBD medication (excluding steroids)		121 (67.60)	54 (60)	57 (72.15)	10 (100)
Any IBD medication or steroids		125 (69.83)	57 (63.33)	58 (73.42)	10 (100)
No medication		27 (15.08)	16 (17.78)	11 (13.92)	0 (0)
Regular IBD medication not recorded		31 (17.32)	20 (22.22)	11 (13.92)	0 (0)

**Table 3 jcm-11-02919-t003:** Mode of delivery.

Method of Delivery (MOD)	IBD	Non-IBD	*p*	RR	95% CI
	N (%)	N (%)			
SVD	97 (54.19)	20,408 (64.73)	0.0033	0.84	0.73–0.96
Breech Vaginal	0 (0)	156 (0.49)	0.35		
Instrumental	28 (15.64)	4405 (13.97)	0.52	1.12	0.80–1.58
CS	54 (30.17)	6559 (20.80)	0.0021	1.45	1.16–1.81
Emergency CS (vs. all MOD)	19 (10.61)	2666 (8.46)	0.30	1.26	0.78–2.07
Emergency CS (vs. elective CS)	19 (35.19)	2666 (40.65)	0.42	0.87	0.60–1.25
Elective CS (vs. non-CS delivery)	35 (21.88)	3893 (13.49)	0.018	1.45	1.08–1.95
Total	179 (100)	31,528 (100)			

**Table 4 jcm-11-02919-t004:** Indications for Caesarean sections.

	IBD Indications, N (%)		Obstetric Indications, N (%)	
Elective	Pouch	5 (35.71)	Breech	11 (52.38)
	Active perianal disease	3 (21.43)	Previous Caesarean	6 (28.57)
	Fistulating disease	3 (21.43)	Unstable lie	2 (9.52)
	Complex surgical history	2 (14.29)	Maternal request	2 (9.52)
	Perforated ileum	1 (7.14)		
	TOTAL	14 (100)	TOTAL	21 (100)
Emergency			Presumed foetal compromise	14 (73.68)
			Undiagnosed breech	2 (10.53)
			Hyperkalaemia	1 (5.26)
			Delay in second stage	1 (5.26)
			Hyperstimulation	1 (5.26)
			TOTAL	19 (100)

**Table 5 jcm-11-02919-t005:** Neonatal outcomes.

Infant Outcomes	IBD	Non-IBD	*p*	RR	95% CI
	N (%)	N (%)			
Preterm	15 (8.52)	2650 (8.47)	0.98	1.01	0.62–1.64
TOTAL	176 (100)	31,297 (100)			
Low birthweight (LBW)	11 (6.15)	2170 (6.90)	0.69	0.89	0.50–1.58
Very low birthweight (VLBW)	3 (1.68)	365 (1.16)	0.52	1.45	0.47–4.46
TOTAL	179 (100)	31,464 (100)			
Small for gestational age (SGA)	13 (8.03)	2617 (8.90)	0.7	0.9	0.54–1.52
TOTAL	162 (100)	29,395 (100)			
NICU	5 (2.96)	1038 (3.52)	0.69	0.84	0.35–2.00
TOTAL	169 (100)	29,457 (100)			
Stillbirth	4 (2.23)	137 (0.44)	0.00031	5.14	1.92–13.75
TOTAL	179 (100)	31,528 (100)			

## Data Availability

Summary data are available on request.

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
