# Peer review of "Risk of Adverse Pregnancy Outcomes for Women with IBD in an Expert IBD Antenatal Clinic"

_jcm, 2022, doi:10.3390/jcm11102919_

Round 1

Reviewer 1 Report

Thank you for the opportunity for reviewing such a quality paper. 

In my opinion, Introduction section is appropriate and provides sufficient data. However, it will be useful to provide data about overall prevalence and incidence regarding IBD in UK and worldwide, if possible. 

From the methodological standpoint, I address no concern. Although analysis was performed retrospectively, I cannot determine any potential bias. 

Results and Discussion section are very well presented - presenting indications among the IBD group is the main strength of this study, and this will certainly contribute to new, prospective studies in this cohort of pregnant women. 

Author Response

see word file

Reviewer 2 Report

Lever et al. present a retrospective study of pregnant women with IBD and their risk of adverse pregnancy outcomes. A cohort of 179 pregnant IBD patients that gave birth at a combined secondary and tertiary clinic was compared to the rest of the pregnant population at the same clinic in the same time period. The main result is that the IBD patients had significantly more planned CS deliveries. There was also a relatively high rate of stillbirths in the IBD group, which was unexpected and difficult to explain.

The women with IBD were recruited from a dedicated combined IBD antenatal clinic. They were older than the control population and included a higher number of primiparous. These factors may contribute to the main result and should be commented in the Discussion.

It is not described whether the IBD antenatal clinic includes all IBD patients from the same area as the control population, or if the IBD antenatal clinic is a referral center for pregnant women with IBD. This should be clarified for readers not familiar with the setting in UK antenatal care. If the clinic is a referral center, the IBD patients may have a more severe disease than the average IBD patient and this should be commented upon.

This study is retrospective. Could this be of influence to the results?

The disease activity is assessed using Physicians global assessment score.  This is not specified. Is no objective disease assessment performed, as a colonoscopy, evaluation of faecal calprotectin, serological markers etc? When was the disease activity assessed? Only once during the pregnancy, or at specified intervals?

Also, there are no data on smoking. Could this influence the result?

Definition on “exposed” to medication during pregnancy: this is not clear to me. Did all the exposed patients receive drugs during at least two trimesters? And all the non-exposed patients were free of the drug from at least 3 months prior to conception? How were patients that received drugs during only one trimester defined?

Table 2: It is difficult to follow the number of patients and the number of missing. What is N and what is % of which N?

Author Response

see word file

Round 2

Reviewer 2 Report

My concerns have been adressed and I have no further comments.